# Antibacterial Effects of Phytocannabinoids

**DOI:** 10.3390/life12091394

**Published:** 2022-09-07

**Authors:** Cassidy Scott, Daniel Neira Agonh, Christian Lehmann

**Affiliations:** 1Department of Pharmacology, Dalhousie University, Halifax, NS B3H 4R2, Canada; 2Department of Anesthesia, Pain Management and Perioperative Medicine, Dalhousie University, Halifax, NS B3H 4R2, Canada

**Keywords:** antibiotics, cannabinoids, infection, inflammation

## Abstract

Antibiotics are used as the first line of treatment for bacterial infections. However, antibiotic resistance poses a significant threat to the future of antibiotics, resulting in increased medical costs, hospital stays, and mortality. New resistance mechanisms are emerging and spreading globally, impeding the success of antibiotics in treating common infectious diseases. Recently, phytocannabinoids have been shown to possess antimicrobial activity on both Gram-negative and Gram-positive bacteria. The therapeutic use of phytocannabinoids presents a unique mechanism of action to overcome existing antibiotic resistance. Future research must be carried out on phytocannabinoids as potential therapeutic agents used as novel treatments against resistant strains of microbes.

## 1. Introduction

Phytocannabinoids are naturally occurring, plant-derived cannabinoids found in some flowering plants, liverworts, and fungi. Phytocannabinoids were first isolated from *Cannabis sativa* (*C. sativa*) and have demonstrated beneficial pharmacological actions for treating pain, inflammation, and infection [1].

*C. sativ**a* is an herbaceous plant that has been used in traditional Asian medicine as a medicinal and psychoactive agent since ancient times [2]. The introduction of cannabis into western medicine occurred during the 19th century. Canada legalised cannabis use for medical purposes in 1999 and for recreational purposes in 2018 [3]. Additionally, in the United States, cannabis is legal in 33 states for medical use and 11 states for recreational use [4]. Cannabis contains several chemically active agents, such as cannabinoids, terpenes, alkaloids, and flavonoids [5]. Phytocannabinoids, which interact with the endocannabinoid system (ECS), exert their function through the activation of cannabinoid receptors one and two (CB1 and CB2) [6].

Antibiotic resistance is a global health challenge, impeding our ability to prevent and treat bacterial infections. New therapeutic methods are constantly sought after for the treatment of bacterial infections. Phytocannabinoids have been shown to possess antibacterial and anti-inflammatory properties, making them a potential target for developing new therapeutics. In this review, we explore the potential therapeutic benefits of phytocannabinoids, and selective CB2 agonists, as treatments for bacterial infections.

## 2. Pharmacology of Antibiotics

Antibiotics are a class of drugs employed against bacterial infections. These drugs work through bacteriostatic (i.e., stopping bacterial reproduction) and bactericidal (i.e., bacterial killing) mechanisms of action. Antimicrobial pharmacotherapy is challenging, as clinicians must select antibiotics that will optimally treat an infection while minimising side effects and the potential development of antimicrobial resistance. Antibiotics are classified based on their mechanism of action, as described in Figure 1.

### 2.1. Antibiotics Targeting Cell Wall Receptors

Bacterial cells are surrounded by a cell wall made of peptidoglycans. The primary targets of β-lactam agents are the penicillin-binding proteins (PBPs). When β-lactam agents are present, they mimic the D-alanyl D-alanine portion of the peptide chain bound by PBPs. Once bound by β-lactam rings, PBPs are not available for the synthesis of new peptidoglycans, resulting in a disruption to the peptidoglycan layer and bacterial lysis. Furthermore, glycopeptides bind to the D-alanyl D-alanine portion of the peptide side chain of precursor peptidoglycan subunits. This prevents binding with PBP and inhibits cell wall synthesis [7].

### 2.2. Inhibition of Protein Biosynthesis

The cytoplasmic ribosomes found in animal cells (80S) differ structurally from those in bacterial cells (70S), making protein biosynthesis a good target for antibacterial drugs. Aminoglycosides are large, highly polar, antibacterial drugs that bind to the 30S subunit of bacterial ribosomes, impairing their proofreading ability and resulting in the production of proteins with incorrect amino acids. These shortened proteins are then inserted into the cytoplasmic membrane, and their disruption leads to bacterial cell death. Aminoglycosides are potent, broad-spectrum drugs and include, for example, streptomycin, gentamycin, neomycin, and kanamycin [8].

Tetracyclines are another class of antibacterial that bind to the 30S subunit. These drugs are bacteriostatic and inhibit protein synthesis by blocking the association of tRNAs with the ribosomes during translation.

Inhibitors of the 50S subunit include macrolides, lincosamides, chloramphenicol, and oxazolidines. Macrolides target the early stages of protein synthesis, namely translocation; they block the elongation of proteins by inhibiting peptide bond formation between specific combinations of amino acids. Lincosamides are similar in their mode of action to macrolides. Chloramphenicol is a structurally distinct class that also binds to the 50S ribosome, inhibiting peptide bond formation. However, chloramphenicol causes anaemia and can cause irreversible loss of blood cell production. Therefore, chloramphenicol is now mainly used in veterinary medicine as its side effects are much less severe in animals. Oxazolidines interfere with the formation of the initiation complex for translation, preventing the translocation of growing proteins from the ribosomal A site to the P site [9].

### 2.3. Inhibition of DNA and RNA Biosynthesis

The enzyme DNA gyrase consists of two A subunits and two B subunits. The A subunit is responsible for cutting double-stranded DNA, the B subunit introduces negative supercoils, and the A subunit then reseals the strands. Fluoroquinolones inhibit DNA gyrase by binding to the A subunit. Moreover, the rifamycin family functions by blocking RNA polymerase activity in bacteria. The RNA polymerase enzymes in bacteria differ structurally from those in eukaryotes, allowing for selective toxicity against bacterial cells [10].

### 2.4. Inhibition of Metabolic Pathways

Drugs that target folic acid synthesis inhibit the enzymes involved in the production of dihydrofolic acid and subsequently pyrimidine and purine nucleic acid synthesis. Sulfonamides inhibit dihydropteroate synthase, while trimethoprim acts at a later stage of folic acid synthesis, inhibiting the enzyme dihydrofolate reductase [8]. Furthermore, isoniazid is another antimetabolite drug that acts by preventing the synthesis of mycolic acid, which is necessary for mycobacterial cell walls [9].

### 2.5. Inhibition of Membrane Function

Polymyxins are lipophilic polypeptide antibiotics that interact with the lipopolysaccharide component of the outer membrane of Gram-negative bacteria. They work by disrupting the outer and inner membranes of bacterial cells. However, polymyxins do not have selective toxicity and can damage cell membranes in the kidneys and the nervous system. Daptomycin works similarly to polymyxins, inserting itself into the bacterial cell membrane of gram-positive bacteria and disrupting it. Its toxicity is better tolerated in humans than polymyxins [11].

### 2.6. Antibacterial Resistance

Many bacterial pathogens associated with human disease have evolved multidrug resistance (MDR) to antibiotic treatment. The emergence of MDR among bacterial pathogens is a major public health threat, jeopardising the ability of clinicians to effectively treat critically ill patients. The Centers for Disease Control and Prevention estimates that more than 2.8 million antibiotic-resistant infections occur in the US each year, and more than 35,000 people die as a result [12].

Bacteria have genetic plasticity that allows them to respond to environmental threats, such as antibiotic molecules. They utilise two main genetic strategies to adapt to antibiotics: (i) mutants in genes associated with the compound’s mechanism of action and (ii) acquisition of foreign DNA, coding resistance through horizontal gene transfer (HGT). Bacterial cells are able to develop mutational resistance when a subset of bacterial cells, derived from a susceptible population, develop genetic mutations that affect the antimicrobial target (i.e., decrease the affinity for the drug), decrease the drug’s uptake, activate mechanisms to expel the molecule, or exert modifications to metabolic pathways. When this resistance occurs, susceptible bacteria will be eliminated, and only resistant bacteria will persist. Acquisition of foreign DNA material through HGT is another important mechanism responsible for the development of resistance. Bacteria are able to acquire external genetic material through transformation, transduction, and conjugation. Resistance in a hospital setting often occurs through conjugation, gene transfer involving cell-to-cell contact, or in the gastrointestinal tract of humans undergoing antibiotic treatment. During conjugation, bacteria use mobile genetic elements (i.e., plasmids and transposons) as vehicles to share genetic information. Finally, site-specific recombination systems, known as integrins, are able to recruit both open reading frames, in the form of mobile gene cassettes, in order to add new genes and the machinery to ensure their expression into bacterial chromosomes [13]. Antibiotic resistance allows for microbes with enhanced morbidity and mortality, due to mutations endowing high levels of antibiotic resistance, increased virulence and enhanced transmissibility [14]. The development of antibiotic resistance poses a major threat. Therefore, novel approaches to treating bacterial infections and overcoming antibiotic resistance are needed.

## 3. The Endocannabinoid System

### 3.1. Endocannabinoid Receptors

The endocannabinoid system consists of endogenous cannabinoids, cannabinoid receptors, and enzymes, which synthesise and degrade endocannabinoids. Endocannabinoids play a role in a variety of physiological and pathophysiological processes. Many of these effects are mediated by two G-protein-coupled receptors (GPCRs), cannabinoid receptors one (CB1) and two (CB2). CB1 receptors are most highly expressed on axons and nerve terminals in the central nervous system (CNS), while CB2 receptors are primarily found in immune cells (Figure 2) [6].

The CB1 receptor is the most abundant GPCR in the brain. It is highly expressed in the basal ganglia nuclei, hippocampus, cerebellum, and cortex. As a result of its distribution within the CNS, CB1 receptor activation plays a role in motor function, cognition, memory, and analgesia [15]. CB1 receptors are present in many presynaptic neurons, mainly located on GABA-ergic axon terminals, in addition to dopaminergic, serotonergic, glutamatergic, and opioid neurotransmitters [16]. CB1 negatively regulates neurotransmitter release by inhibiting the release of excitatory and inhibitory neurotransmitters via inhibiting the phosphorylation of A-type potassium channels. Additionally, CB1 inhibits N-type calcium channels through direct interaction with the inhibitory G-protein [17].

The CB2 receptor is highly expressed peripherally on immune cells and in tissues of the immune system, including the spleen, tonsils, and thymus [15]. Moreover, B lymphocytes have been shown to express the highest amounts of CB2 [18]. CB2 receptors have been shown to be up-regulated in response to immune cell activation and inflammation [17]. CB2 receptors exert their effects through the initiation of phospholipase C (PLC), inositol 1, 4, and 5-triphosphate (IP3) signalling pathways that lead to increased levels of intracellular calcium [19].

Cannabinoid receptors are those receptors that respond to cannabinoid drugs, such as Δ9-tetrahydrocannabinol (THC), derived from *Cannabis sativa*. Many therapeutics used to target endocannabinoid systems are derived from *C. sativa* and produce their effects by activation of cannabinoid receptors. However, the psychoactive effects of these compounds have restricted their potential application in western medicine.

### 3.2. Cannabinoid Ligands and Enzyme Inhibitors

There are five structurally distinct classes of cannabinoid compounds: classical cannabinoids (e.g., Δ9-THC, Δ8-THC, synthetic Δ8-THC analogue (HU210)); bicyclic cannabinoids (e.g., CP-55,940); indole-derived cannabinoids (e.g., WIN 55212-2); eicosanoids (e.g., anandamide, 2-AG); and antagonist/inverse agonists (e.g., SR141716A for CB1, SR145528 for CB2) [15]. Some agonists show little selectivity between CB1 and CB2 receptors. However, antagonist compounds are usually highly selective, allowing for discrimination between CB1- and CB2-mediated effects. The classical group consists of dibenzopyran derivates. Δ9-THC is the main psychoactive constituent of cannabis. HU-210 is a synthetic analogue of Δ8-THC, with a higher affinity for CB1 and CB2 receptors, a higher potency, and a higher relative intrinsic activity as a cannabinoid receptor agonist than Δ9-THC. The bicyclic analogues of Δ9-THC lack a pyran ring. CP55,940 is a well-known member of this group and has been found to have slightly lower CB1 and CB2 affinities compared to HU-210. Indole-derived cannabinoids have structures that differ markedly from the previously mentioned groups. WIN 55212-2 is the most studied indole-derived cannabinoid. WIN 55212-2 is similar to HU-210 and CP55,940 as it has intrinsic activity at both CB1 and CB2 receptors. However, unlike these cannabinoids, it has been found to have a slightly higher affinity for CB2 than CB1. The two main endocannabinoids in the eicosanoid group are N-arachidonoylethanolamide (anandamide) and 2-arachidonoylglycerol (2-AG). These are the primary endogenous ligands that bind and activate CB1 and CB2. Anandamide and 2-AG are both synthesised on demand, in response to elevations of intracellular calcium [20].

Anandamide is rapidly hydrolysed in the CNS by fatty acid amide hydrolase (FAAH) [21]. Additionally, anandamide can be degraded via oxidation by cyclooxygenase-2 (COX-2). COX-2 is reasonably selective for anandamide over other acyl ethanol amides, so its inhibition allows for a more selective way to increase anandamide concentrations. Lastly, anandamide degradation can occur via N-acylethanolamine-hydrolysing acid amidase (NAAA) [22].

2-AG degradation is primarily due to monoacylglycerol lipase (MAGL) and alpha/beta domain hydrolases 6 and 12 (ABHD6 and 12). Enzymes that degrade 2-AG have different subcellular localisation, which allows for the degradation of 2-AG in different cellular compartments. MAGL is widespread within the CNS and accounts for the majority of the 2-AG degradation in the brain. ABDH6 is primarily localised to dendrites and dendritic spines of excitatory neurons in the cortex. Lastly, ABDH12 is involved in the hydrolysis of 2-AG in the brain (Figure 3).

## 4. Antibacterial Effects of Cannabinoids

### 4.1. Classification of Phytocannabinoids

Phytocannabinoids are meroterpenoids containing a resorcinol core with an isoprenyl, alkyl, or aralkyl para-positioned side chain. The first phytocannabinoids were isolated from *C. sativa,* and to date, over 113 phytocannabinoids have been isolated from the plant [23]. Cannabinoids are classified into distinct types: cannabigerols (CBGs), cannabichromenes (CBCs), cannabidiols (CBDs), (-)-Δ^9^-*trans*-tetrahydrocannabinols (Δ^9^-THCs), (-)-Δ^8^-*trans*-tetrahydrocannabinols (Δ^8^-THCs), cannabicyclols (CBLs), cannabielsoins (CBEs), cannabinols (CBNs), cannabinodiols (CBNDs), cannabitriols (CBTs), and the miscellaneous cannabinoids (Figure 4) [1].

In addition to phytocannabinoids synthesised from cannabis, several *Rhododendron* species produce cannabinoids, typically belonging to the CBC type [24]. Flowering plants such as *Helichrysum umbraculigerum* Less., *Glycyrrhiza foetida* (liquorice), and *Amorpha fruticosa* (indigo) contain bioactive compounds with a cannabinoid backbone, carrying an aralkyl side chain [25,26]. Cannabinoids, including bibenzyl analogues of Δ9-THC, have been isolated from liverworts, including *Radula perrottetii* [27]. Lastly, cannabinoids have been isolated from fungal organisms, including Cannabiorcichromenic acid from *Cylindrocarpon oidium* [28].

### 4.2. Antibacterial Effects of Cannabidiol (CBD)

The treatment of bacterial infections is becoming increasingly complicated due to the emergence of antibiotic resistance. Recently, there has been extensive research done on the antimicrobial activity of cannabinoids (Table 1).

Studies by Wassmann et al. (2020) demonstrated the antimicrobial effects of CBD against Gram-positive bacteria when used in combination with the cyclic peptide antibiotic bacitracin (BAC). CBD reduced the minimum inhibitory concentration (MIC) of BAC by at least 64-fold and was effective against *Staphylococcus* species, *Listeria monocytogenes,* and *Enterococcus faecalis*. Although the antimicrobial mechanisms of CBD remain largely unknown, this study showed CBD and BAC combination induced several septa formations during cell division, along with membrane irregularities, suggesting that CBD may exert antimicrobial activity through mechanisms affecting the cell envelope [29].

Studies by Blaskovich et al. (2021) confirmed reports of CBD treatment against Gram-positive bacteria, including the highly resistant *Staphylococcus aureus*, *Streptococcus pneumoniae*, and *Clostridioides difficile,* while demonstrating that CBD can also selectively kill a subset of Gram-negative bacteria, including *Neisseria gonorrhoeae*. CBD activity against Gram-negative bacteria is limited due to the presence of the outer membrane and lipopolysaccharides (LPS). Therefore, when CBD was used in combination with membrane-disrupting drugs, CBD susceptibility increased. Future studies should aim to evaluate the variations in outer membrane compositions that affect CBD antibacterial activity [30].

### 4.3. Antibacterial Effects of Cannabichromene (CBC)

Studies by Stahl et al. (2020) examined the efficacy of different cannabinoids in reducing bacterial contents from dental plaque. They found CBC and cannabinol (CBN) to be the most effective for reducing bacterial growth, while cannabigerolic acid (CBGA), cannabigerol (CBG), and cannabidiolic acid (CBDA) all had significantly lower colony counts when compared to toothpaste [31]. Furthermore, cannabichromenic acid (CBCA), the chemical precursor of CBC, has been studied for its antibacterial activity against methicillin-resistant *Staphylococcus aureus*. Galletta et al. (2020) demonstrated that CBCA had faster, more potent bactericidal activity against methicillin-resistant *Staphylococcus aureus* (MRSA) when compared to vancomycin. Their studies showed that CBCA might exert antibacterial effects by degrading the bacterial lipid membrane and altering the bacterial nucleoid [32].

### 4.4. Antibacterial Effects of Cannabigerol (CBG)

Studies by Farha et al. (2020) demonstrated the antibacterial activity of cannabigerol (CBG) in a murine model of systemic infection caused by MRSA. Results of their study revealed that CBG works by targeting the cytoplasmic membrane of Gram-positive bacteria, inhibiting its ability to form biofilms, therefore eradicating preformed biofilms. Furthermore, CBG was able to work in combination with polymyxin B against multidrug-resistant Gram-negative pathogens [33]. These results demonstrated the potential for cannabinoids to work on Gram-negative organisms whose outer membrane were permeabilised. Additionally, studies by Aqawi et al. (2021) demonstrated the antibacterial activity of CBG against *Streptococcus mutans.* CBG was able to halt the proliferation of planktonic growing *S. mutans* and alter membrane structure. Treatment with CBG led to intracellular accumulation of membrane structures, induced membrane hyperpolarisation, decreased membrane fluidity, and prevented the *S. mutans*-induced drop in pH. Together, these results suggested potential antibacterial mechanisms of CBG against *S. mutans* [34].

### 4.5. Antibacterial Effects of Cannabidiolic Acid (CBDA)

Studies by Martinenghi et al. (2020) looked at the antimicrobial effects of CBDA and its decarboxylated form CBD. Both compounds showed significant antimicrobial effects on Gram-positive *S. aureus* and *S. epidermidis*, but no activity was seen on Gram-negative *Escherichia coli* or *Pseudomonas aeruginosa*. CBDA presented a two-fold lower antimicrobial activity than its decarboxylated form. The authors rationalised this by suggesting that the antimicrobial activity of cannabinoids was related to their ability to permeabilise the bacterial cell membrane, acting as a detergent-like agent [35].

### 4.6. Antibacterial Effects of Other CBR Agonists

In addition to phytocannabinoids, natural CB2 agonists are of interest due to their anti-inflammatory and antioxidant activity [36]. β-caryophyllene is a selective CB2 agonist found in the essential oils of edible plants such as *Cannabis sativa*, cloves, oregano, and cinnamon; it is a dietary cannabinoid with a generally recognised as safe (GRAS) status, approved by the FDA for food use. This substance is of particular interest due to its low toxicity, local anaesthetic activity [37], anti-inflammatory capacity [38], and antibacterial action. Studies by Woo et al. (2020) demonstrated the antibacterial effects of β-caryophyllene against *Helicobacter pylori.* In vitro, the β-caryophyllene treatment decreased the expression of the *H. pylori* replication genes necessary for cell growth. Furthermore, β-caryophyllene successfully eradicated *H. pylori* infection in Mongolian gerbils and diminished inflammation in infected stomach tissues [39]. Similar studies by Jung et al. (2020) supported these results, demonstrating that *H. pylori* infection levels and gastric mucosal inflammation decreased dose-dependently after β-caryophyllene treatment [40].

Studies by Pieri et al. (2016) evaluated the antimicrobial activity of β-caryophyllene against bacteria from a dog’s dental plaque. β-caryophyllene inhibited the adherence capability of *Lactococcus*, *Streptococcus* and *Bacillus* isolates in vitro. Additional, β-caryophyllene inhibited dental plaque formation in dogs in vivo by 37.5% [41].

Lastly, studies by Iseppi et al. (2019) examined the antibacterial activity of essential oils from Fibre-Type Cannabis sativa L. (Hemp). Their results demonstrated that the main components of hemp essential oils, β-caryophyllene, α- and β-pinene, and β-myrcene provided antibacterial activity against Gram-positive bacteria. All components exhibited antibacterial activity against *Staphylococcus*, *Listeria*, *Enterococcus*, and *Bacillus* strains in vitro, expressed by the inhibition of growth [42].

Additionally, the bibenzylic cannabinoid amorfrutin has demonstrated antibacterial activity in Gram-positive and acid-fast bacteria. Studies by Mitscher et al. (1981) isolated amorfrutin A and B from *Amorpha fruticosa*. Both amorfrutin A and B demonstrated antimicrobial activity in vitro against *Staphylococcus aureus* and *Mycobacterium smegmatis* [43].

Lastly, extracts of the *Rhododendron* species have demonstrated antibacterial activity against different Gram-positive and Gram-negative bacteria. Studies by Said et al., 2017 demonstrated the antibacterial activity of Cannabiorcichromenic acid (CCA), derived from *Rhododendron collettianum*, against *B. thioparus*, *B. aquimaris*, *B. subtillis*, *Staphylococcus epidermidis*, and *E. coli* [44].

**Table 1 life-12-01394-t001:** Minimum inhibitory concentrations [MIC, (in µg/mL)] of antibacterial activity of different phytocannabinoids against various bacteria.

Strain	CBD	CBCA	CBG	THC	CBDA	BCP
*Staphylococcus aureus* (MRSA)	4 2	3.9	2	2	4	16
*Enterococcus faecalis*	8 1	7.8				1
*Listeria monocytogenes*	4 1					2
*Streptococcus pneumoniae*	1–2					6250
*Clostridioides difficile*	2–4					
*Staphylococcus epidermidis*	2				4	
*Bacillus subtilis*	8					1

Based on [29,30,32,33,35,41,42].

### 4.7. The Potential Role of Cannabinoid Structure on Antibacterial Activity

The molecular mechanisms behind the antibacterial activity of cannabinoids have yet to be fully elucidated. Studies by Appendino et al. (2008) examined the effects of structural modification on the bactericidal activity of five major cannabinoids (CBD, CBC, CGB, THC, and CBN). All five cannabinoids demonstrated potent activity against a variety of MRSA strains, with MIC values between 0.5 and 2 µg/mL. Methylation and acetylation of the phenolic hydroxyls, esterification of the carboxylic group of cannabinoids, and the introduction of a second prenyl moiety were detrimental to the cannabinoids’ antibacterial activity. Whereas antibacterial activity was tolerant to the nature of the prenyl moiety, its relative position compared to the *n*-pentyl moiety (abnormal cannabinoids), and the carboxylation of the resorcinol moiety (pre-cannabinoids). The authors rationalised that these results demonstrated tolerance to the structural modification of the terpenoid moiety, suggesting that these residues serve mainly as modulators of lipid affinity, while the addition of further prenyl moiety may result in poorer aqueous solubility, leading to a loss of antibacterial activity [45].

## 5. Therapeutic Applications

Cannabinoids can be used therapeutically via different routes of administration, most commonly through inhalation and oromucosal delivery. When administered via inhalation, peak plasma concentrations are attained rapidly, within 3–10 min, and the maximum concentrations are higher relative to when taken orally. THC and CBD are both highly lipophilic and have poor bioavailability; additionally, an extensive hepatic first-pass metabolism further reduces bioavailability. Therefore, oral administration is slow, and maximal plasma concentrations can be observed as late as 4–6 h [46]. Using a vaporiser to administer cannabinoids is another method with similar pharmacokinetics to smoked cannabinoids but avoids the respiratory risk associated with smoking. Additionally, transdermal administration of cannabinoids can be used to avoid first-pass metabolism, but the highly hydrophobic nature of cannabinoids limits diffusion across the aqueous layer of skin. Permeation enhancement can be used to overcome this limitation [46]. Intravenous (IV) administration of cannabinoids is not currently used in humans but has been studied in rodent models. Results have shown IV administration to overcome the drawbacks of both inhalation and oral administration (i.e., respiratory risks and first-pass metabolism, respectively), offering a potential benefit to the development of IV formulation [47,48].

## 6. Conclusions

Current antibiotic treatments have limited efficacy against multidrug-resistant bacteria, causing a significant challenge for prescribing physicians. A lack of effective therapies or new antibiotics requires the development of alternative antimicrobial therapies. Research has shown phytocannabinoids and CB2 agonists to exhibit antibiotic activity against a variety of Gram-positive and Gram-negative bacteria. Although their antimicrobial activity is limited in terms of Gram-negative bacteria, they offer therapeutic potential when administered as an adjunct treatment with an outer membrane perturbing molecule to facilitate the permeation of compounds that are effective on Gram-positive bacteria. Research has also shown synergy supporting the potential for combination therapy both in vivo and in vitro. Furthermore, CB2 agonists, such as β-caryophyllene, are widely used in industry as food additives and traditional medicine, and many are FDA approved and generally recognised as safe (GRAS), making them a good option for a novel therapeutic. The studies presented in this review suggest an attractive potential for cannabinoid-based antibacterial treatments. However, future research further detailing the mechanisms of action against bacteria is necessary.

## Figures and Tables

**Figure 1 life-12-01394-f001:**
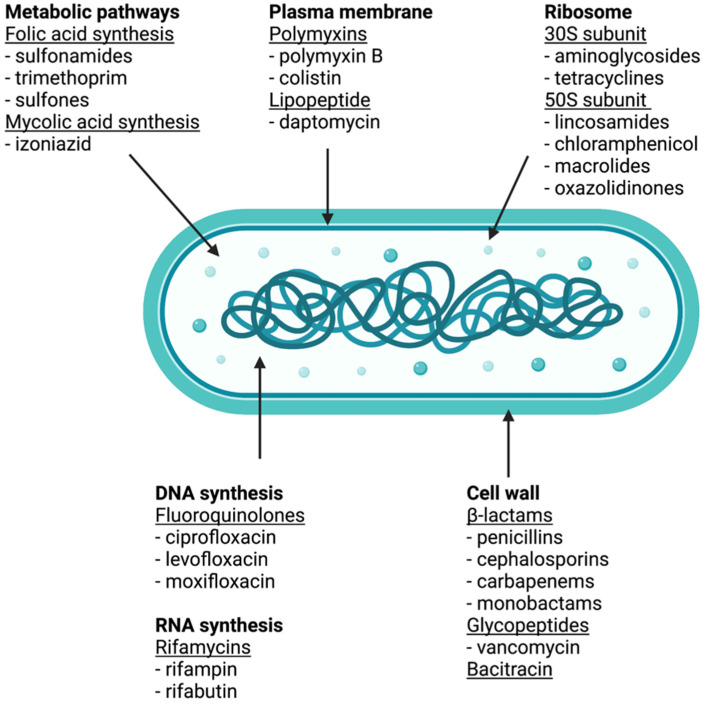
Classes of antibacterial compounds based on their bacterial target. Figure created with BioRender.com.

**Figure 2 life-12-01394-f002:**
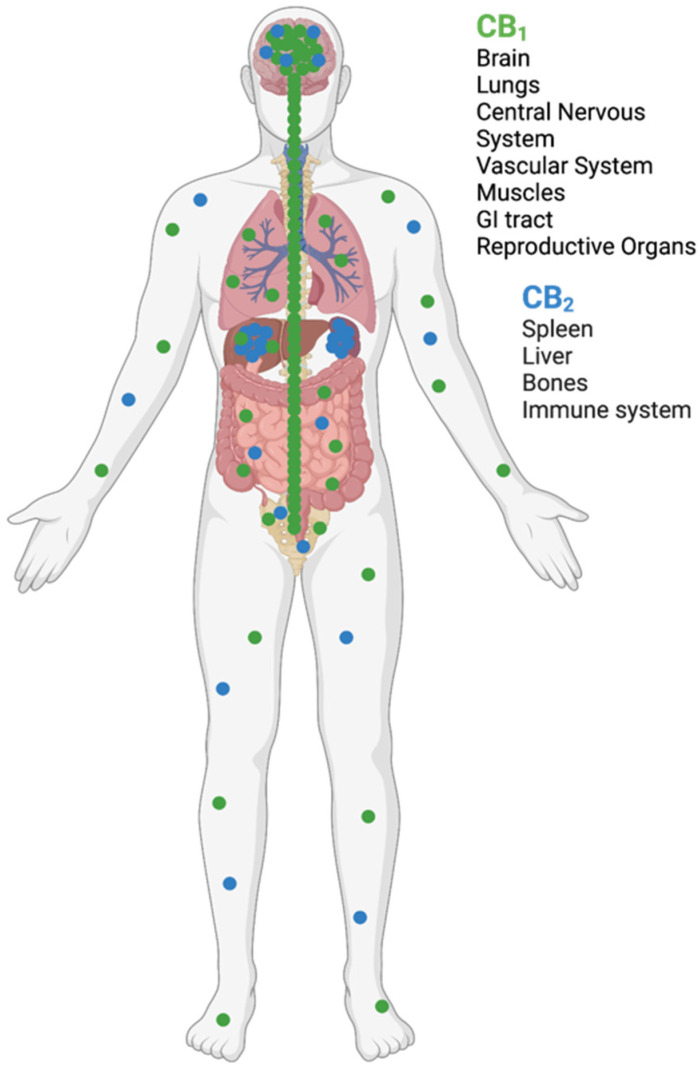
Distribution of cannabinoid receptors one (CB1) and two (CB2) in the human body. Figure created with BioRender.com.

**Figure 3 life-12-01394-f003:**
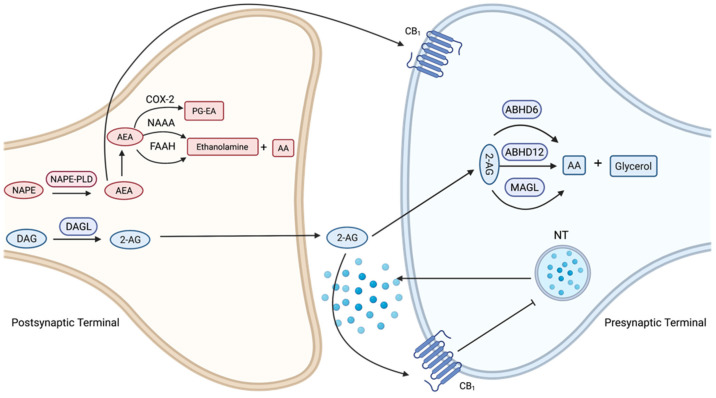
Simplified schematics of endocannabinoid retrograde signalling-mediated synaptic transmission. The primary endogenous ligands that bind and activate cannabinoid receptors one (CB1) and two (CB2) are N-arachidonoylethanolamide (anandamide) and 2-arachidonoylglycerol (2-AG). Anandamide (AEA) is biosynthesised by N-acyl-phosphatidylethanolamine (NAPE) by NAPE-specific phospholipase D (NAPE-PLD), and 2-AG is biosynthesised by diacylglycerol (DAG) by diacylglycerol lipase-α (DAGLα). Endocannabinoids readily cross the membrane and travel in a retrograde manner to activate CB1 receptors in the presynaptic terminal. Activated CB1 receptors inhibit neurotransmitter (NT) release through the suppression of calcium influx. Anandamide is hydrolysed by fatty acid amide hydrolase (FAAH) and can be degraded via oxidation, by cyclooxygenase-2 (COX-2) or via N-acylethanolamine-hydrolysing acid amidase (NAAA). 2-AG degradation is primarily due to monoacylglycerol lipase (MAGL) and alpha/beta domain hydrolases 6 and 12 (ABHD6 and 12). Figure created with BioRender.com.

**Figure 4 life-12-01394-f004:**
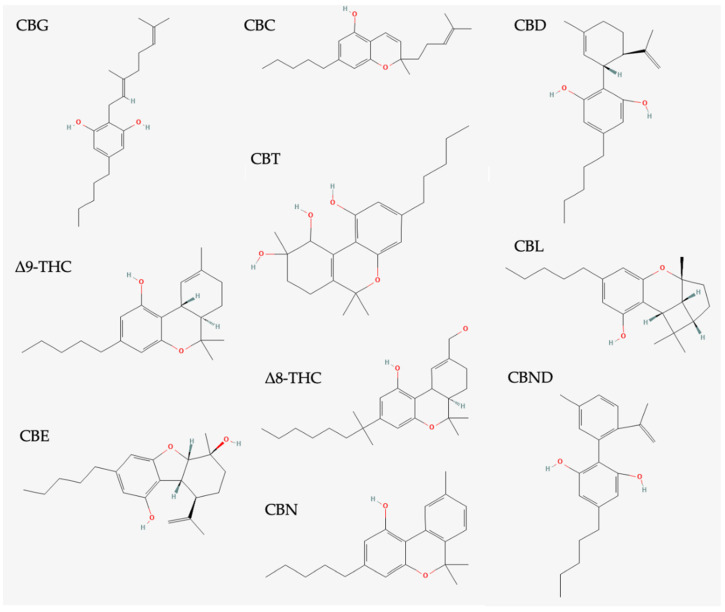
Overview of cannabinoid structures. Cannabigerol (CBG), cannabichromene (CBC), cannabidiol (CBD), (-)-Δ^9^-*trans*-tetrahydrocannabinol (Δ^9^-THC), cannabitriols (CBTs), cannabicyclol (CBL), cannabielsoin (CBE), (-)-Δ^8^-*trans*-tetrahydrocannabinol (Δ^8^-THC), cannabinodiol (CBND), and cannabinol (CBN).

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
