# Peer review of "Antibacterial Effects of Phytocannabinoids"

_life, 2022, doi:10.3390/life12091394_

Round 1

Reviewer 1 Report

I would recommend writing  „Phytocannabinoids” instead of „Phyto-Cannabinoids”

I am confused by the use of the word “cannabinoid” and I think that this should be revised through the manuscript e.g. what means “Non-Cannabinoid CB2 Agonists” in my view CB2 agonists are cannabinoid ligands, ligands acting at the cannabinoid receptors type 2.

The same for “natural non-cannabinoid CB2 agonists”

For β-caryophyllene the authors are saying “It is a dietary cannabinoid” which is a contradiction, β-caryophyllene is non-cannabinoid but a cannabinoid.

Line 337, is this sentence correct: “The studies presented in this review suggest an attractive potential for antibiotic drugs.”

Author Response

Thank you for your availability and comments on this review article. 

Point 1: I would recommend writing “Phytocannabinoids” instead of “Phyto-cannabinoids”  

Response: “Phyto-cannabinoids” written in the title, introduction, and section 5.5, were changed to phytocannabinoids.  

Point 2: I am confused on the use of the word “cannabinoid” and I think this should be revised through the manuscript e.g. what means “Non-cannabinoid CB2 agonists” in my view CB2 agonists are cannabinoid ligands, ligands acting at the cannabinoid receptors type 2.

Response: Thank you for this comment, Section 5.5 was revised. ‘Non-cannabinoid CB2 agonists’ was changed to “Other CBR agonists”.

Point 3: The same for “natural non-cannabinoid CB2 agonists”

Response: “Natural non-cannabinoid CB2 agonists” was changed to “natural CB2 agonists.”.  

Point 4: “For ?-caryophyllene, the authors are saying “it is a dietary cannabinoid” which is a contradiction, ?-caryophyllene is a non-cannabinoid but a cannabinoid”. 

Response: Thank you, as highlighted in point 2 we removed the use of “non-cannabinoid” and the sentence now reads “it is a dietary CB2 agonist”

Point 5: “Line 337, is this sentence correct: “The studies presented in this review suggest an attractive potential for antibiotic drugs.” 

Response: Thank you, this has been revised and “The studies presented in this review suggest an attractive potential for antibiotic drugs” was changed to “The studies presented in this review suggest an attractive potential for cannabinoid based antibacterial treatments.” 

Reviewer 2 Report

Hi Authors, I would like to thank you for this kind of review and I felt that the contents are not enough to justify your review. The title is antibacterial activity of phytocannabinoids, so you should collect the vast literature review to justify the title. I found article titled as "Antibacterial Cannabinoids from Cannabis sativa: A Structure-Activity Study" published in J. Natural Products 2008. This article is missing in your review. Collect the articles about the phytocannabinoids other than Cannabis sativa, like Glycyrrhiza foedita, Amorpha fructosa, Radulla marginata and many other plants, include in your review. You should present the chemical classification of phytocannabinoids with structures, i.e acidic cannabinoids and its decarboxylated derivatives of neutral cannabinoids or different types of phytocannabinoids like CBC, CBD, CBE, CBG, CBL, CBN, CBND, CBT and THC type. Please collect sufficient number of articles related to antibacterial activity and present in the tabular form including MIC or inhibitory concentration. 

Herewith I enclosed the spelling mistakes of the review. Thank you.

Author Response

Thank you for your availability and comments on this review article. 

Point 1: I found article titled as “Antibacterial Cannabinoids from Cannabis sativa: A Structure-Activity Study” published in J. Natural Products 2008. This article is missing in your review. 

Response: Thank you very much for your suggestion – we indeed missed this paper. We now included a new paragraph in our manuscript discussing this publication (Section 4.7.): 

Molecular mechanisms behind the antibacterial activity of cannabinoids have yet to be fully elucidated. Studies by Appendino et al. (2008) examined the effects of structural modification on bactericidal activity of five major cannabinoids (CBD, CBC, CGB, THC and CBN). All five cannabinoids demonstrated potent activity against a variety of MRSA strains with MIC values between 0.5-2 ug/ml. Methylation and acetylation of the phenolic hydroxyls, esterification of the carboxylic group of pre-cannabinoids and introduction of a second prenyl moiety were detrimental to the cannabinoid’s anti-bacterial activity. Whereas anti-bacterial activity was tolerant to the nature of the prenyl moiety, its relative position compared to the n-pently moiety (abnormal cannabinoids) and carboxylation of the resorcinyl moiety (pre-cannabinoids). The authors rationalize that these results demonstrating tolerance to structural modification of the terpenoid moiety suggest that these residues serve mainly as modulators of lipid affinity while addition of further prenyl moiety may result in poorer aqueous solubility, leading to a loss of antibacterial activity.

Point 2: Collect the articles about the phytocannabinoids other than Cannabis sativa, like Glycyrrhza doedita, Amorpha fructosa, Radulla marginata, and many others plants, include in your review.

Response: Thank you for this suggestion. A new section entitled “4.1. Classification of Phytocannabinoids” has been added. This section details phytocannabinoids isolated from species other than cannabis including flowering plants, liverworts and fungal organisms. Additionally, section “4.6. Antibacterial Effects of Other CBR agonists” has been expanded to include non-cannabis-based cannabinoids. Although there are other plants containing phytocannabinoids, the link between their cannabinoid components and their antibacterial effects is often not studied.

Point 3: You should present the chemical classification of phytocannabinoids with structures, i.e., acidic cannabinoids and its decarboxylated derivatives of neutral cannabinoids or different types of phytocannabinoids like CBC, CBD, CBE, CBG, CBL, CBN, CBND, CBT, and THC type. 

Response: Thank you for this comment. Figure 4 has been added which includes the chemical structures of different cannabinoids presented in our paper.

Point 4: Please collect sufficient number of articles related to antibacterial activity and present in tabular form including MIC or inhibitory concentration.

Response: Table 1 consisting of MIC of antibacterial activity of different phytocannabinoids against various bacteria was made and added to the review.

Additional Comments: Thank you for your English corrections, all changes have been made to the manuscript.

Round 2

Reviewer 2 Report

Hi author(s), I would like to thank you for addressing my suggestions. The Table-1 mentions some numbers like 4 [29], what is the meaning? Is it MIC and reference number? If it is then please mention below the table. The space between the words Bacillus subtilis

Author Response

Comment 1: The Table-1 mentions some numbers like 4 [29], what is the meaning? Is it MIC and reference number? If it is then please mention below the table. 

Response: Thank you for your additional comment. The number in square brackets was used to represent the reference. We have moved the references to below the table.  

Comment 2: The space between the words Bacillus subtilis  

Response: Thank you for this correction. A space has been added in Bacillus subtilis